# OpenReview forum: "Towards LLM Unlearning Resilient to Relearning Attacks: A Sharpness-Aware Minimization Perspective and Beyond"
_ICML.cc/2025/Conference — ICML 2025 poster_

### Official Review · Reviewer_s57J · 2025-03-10

**Overall Recommendation:** 3

**Summary:**

This paper explores the robustness of large language model (LLM) unlearning against relearning attacks, which can effectively restore forgotten knowledge through minimal fine-tuning. The authors establish a connection between robust LLM unlearning and Sharpness-Aware Minimization (SAM), a technique designed to improve generalization by flattening the loss landscape.

The contributions of this paper include:

1. Formulating LLM unlearning as a min-max optimization problem, analogous to adversarial training, where the adversary aims to reverse the unlearning effect.

2. Demonstrating that SAM and broader smoothness optimization techniques (gradient penalty, curvature regularization, randomized smoothing, weight averaging) enhance robustness against relearning attacks.

3. Conducting extensive experiments on WMDP and MUSE datasets, showing that smoothness optimization significantly improves LLM unlearning stability.

4. Extending the framework to defend against jailbreaking attacks, making LLM unlearning more resistant to adversarial prompting

**Claims And Evidence:**

yes

**Essential References Not Discussed:**

see Methods And Evaluation Criteria

**Ethical Review Concerns:**

no need

**Experimental Designs Or Analyses:**

see Methods And Evaluation Criteria

**Methods And Evaluation Criteria:**

pros
1. The analogy between unlearning robustness and adversarial training is insightful and well-justified. Introducing SAM-based optimization as a solution is a novel contribution to the field.

2. Conducts large-scale experiments across two benchmarks (WMDP, MUSE) and multiple unlearning techniques. Evaluates both relearning attacks (fine-tuning-based) and jailbreaking attacks (prompt-based), covering diverse adversarial settings.

3. Shows that SAM-enhanced unlearning consistently outperforms standard methods. Provides quantitative insights into how different smoothness optimization techniques (RS, GP, CR, WA) impact unlearning resilience.

4. Provides min-max optimization analysis to justify why sharpness-aware methods improve unlearning stability. Derives the connection between curvature regularization and robustness against relearning.

Cons:
1. The paper only considers small-scale fine-tuning-based relearning attacks. More adaptive attack strategies, such as gradient inversion e.g., [1] or meta-learning-based attacks, e.g., [2], should be explored.

2. SAM and second-order smoothness techniques introduce non-trivial training costs, which might limit practical adoption in large-scale  LLMs (e.g., GPT-4, PaLM). Discussion on efficiency trade-offs is missing—how does increased robustness affect model training time?

3. While WMDP and MUSE are useful benchmarks, real-world regulatory or compliance-driven unlearning cases (e.g., GDPR data deletion) should be considered. Scalability beyond benchmark datasets is not addressed—how does the method perform in internet-scale training corpora?

4. Jailbreaking defenses are briefly discussed, but more sophisticated adaptive jailbreak attacks (e.g., prompt engineering-based attacks) should be tested. Evaluating how well unlearned models withstand iterative adversarial prompting would strengthen the claim that smoothness optimization mitigates jailbreaking.

[1] Zhang, Rui, et al. "A survey on gradient inversion: Attacks, defenses and future directions." arXiv preprint arXiv:2206.07284 (2022).

[2] Gong, Xueluan, et al. "Augmenting Model Extraction Attacks Against Disruption-Based Defenses." IEEE Transactions on Information Forensics and Security (2024).

**Other Comments Or Suggestions:**

see Methods And Evaluation Criteria

**Other Strengths And Weaknesses:**

see Methods And Evaluation Criteria

**Questions For Authors:**

see Methods And Evaluation Criteria

**Relation To Broader Scientific Literature:**

see Methods And Evaluation Criteria

**Theoretical Claims:**

no proof

---

> ### Author Rebuttal · Authors · 2025-03-31
>
> We appreciate Reviewer s57’s careful evaluation of our work. The constructive criticism and insightful questions help us further improve the paper. We respond to each key question below.
>
> 1. **Response to the choice of attacks**
>
> Thank you for raising this question. Based on your suggestion, we have looked into the suggested references and their attack methods. However, we feel that they were not very appropriate in the context of this work.
>
> We choose relearning attacks and jailbreaking attacks as the main evaluation settings in this work because they are widely recognized as the two dominant and SOTA attack types in the LLM unlearning literature [1][2][3]. These attacks align well with the general machine unlearning setting, where an attacker either fine-tunes the unlearned model using a small amount of data or uses adversarial prompts to circumvent unlearning effects. However, gradient inversion and meta-learning-based attacks are primarily developed for CNN-based image classification models and thus, it is  difficult to directly adapt these attacks to LLM unlearning.
>
>
> 2. **Response to computation efficiency and larger models**
>
> Thank you for your insightful suggestion. **[Fig. R1](https://ibb.co/4Rqfnq8d)** presents the total run time of our proposed smoothness-enhanced unlearning approachs as well as the additional baseline approach, **TAR** (Tampering Attack Resistance via meta-learning) [1]. As we can see, NPO+SAM shows the second-best efficiency, slightly behind NPO+WA. It approximately doubles the runtime of vanilla NPO due to the one-step maximization for weight perturbation in the alternating gradient ascent-descent implementation of SAM (Appendix A). In addition, TAR incurs a prohibitively high computational cost, with a running time of 7,441.9 minutes, which is 647× slower than NPO+SAM.
>
> In addition, **[Table R1](https://ibb.co/S7fFVRfX)** provides additional robustness comparison results using a **larger model, LLaMA 3 8B**, which is the largest model supported in our lab environment. In addition to TAR, we also considered another baseline approach **LAT** (Latent Adversarial Training, which enhances robustness to persistent harmful behaviors in LLMs by adding perturbations to neuron activations) [5]. As we can see, NPO+SAM delivers highly competitive performance on the WMDP-Bio unlearning task, comparable to TAR and significantly outperforming LAT. This shows the consistent effectiveness of NPO+SAM on the larger 8B-sized model.
>
> 3. **Response to benchmark choice**
>
> To the best of our knowledge, WMDP is a representative benchmark closely aligned with practical unlearning needs, focusing on the removal of harmful or sensitive knowledge (e.g., biological facts) from pre-trained LLMs. MUSE is another widely used benchmark for data- and knowledge-wise unlearning, including copyrighted books (MUSE-Books) and real-world news (MUSE-News). These benchmarks reflect real-world goals: WMDP promotes safe content generation, while MUSE addresses copyright concerns.  Moreover, we are not aware of any established unlearning benchmark built on internet-scale training corpora. This is expected, as unlearning is fundamentally different from pretraining, and typically operates under a well-defined, narrow unlearning scope, as evidenced by the small size of forget datasets in existing benchmarks.
>
> 4. **Response to the choice of jailbreaking attack**
>
> Thank you for raising this question. To clarify, Enhanced-GCG is a prompt-engineering-based, adaptive attack that optimizes prompts against the unlearned model, making it particularly effective at bypassing unlearning defenses. We selected Enhanced-GCG because it has been shown to be the most effective jailbreaking attack specifically designed for LLM unlearning [3]. In contrast, other jailbreaking methods [6][7] fail to reliably compromise unlearned models, rendering them less suitable for evaluating worst-case robustness of LLM unlearning against input-level attack.
>
> [1] Lynch, Aengus, et al. "Eight methods to evaluate robust unlearning in LLMs." arXiv preprint arXiv:2402.16835 (2024).
>
> [2] Hu, Shengyuan, et al. "Jogging the Memory of Unlearned Models Through Targeted Relearning Attacks." ICML 2024 Workshop on Foundation Models in the Wild.
>
> [3] Łucki, Jakub, et al. "An adversarial perspective on machine unlearning for AI safety." arXiv preprint arXiv:2409.18025 (2024).
>
> [4] Tamirisa, Rishub, et al. "Tamper-resistant safeguards for open-weight llms." arXiv preprint arXiv:2408.00761 (2024).
>
> [5] Sheshadri, Abhay, et al. "Latent adversarial training improves robustness to persistent harmful behaviors in llms." arXiv preprint arXiv:2407.15549 (2024).
>
> [6] Li, Nathaniel, et al. "The WMDP benchmark: Measuring and reducing malicious use with unlearning." arXiv preprint arXiv:2403.03218 (2024).
>
> [7] Huu-Tien, Dang, et al. "On effects of steering latent representation for large language model unlearning." arXiv preprint arXiv:2408.06223 (2024).

---

### Official Review · Reviewer_hvpj · 2025-03-12

**Overall Recommendation:** 3

**Summary:**

The paper reveals that Sharpness-Aware Minimization (SAM), traditionally used for improving model generalization, naturally yields a robust optimization framework for LLM unlearning. Through experiments, the paper shows that SAM-enhanced unlearning methods result in smaller discrepancies between model performance before and after relearning attacks, indicating better retention of unlearned information.

**Claims And Evidence:**

The paper presents experimental results and visualizations that support its claims and highlights the potential of SAM as a tool for improving the security and privacy of LLMs.

**Essential References Not Discussed:**

No

**Experimental Designs Or Analyses:**

The experimental design involves evaluating the unlearning robustness of the methods on different datasets, such as WMDP, AGNews, GSM8K, and SST2, under various relearning attack settings. The paper also reports utility performance results to assess the balance between unlearning and model performance preservation. No obvious issues with the experimental design or analyses were apparent.

**Methods And Evaluation Criteria:**

Yes

**Other Comments Or Suggestions:**

None.

**Other Strengths And Weaknesses:**

None.

**Questions For Authors:**

None.

**Relation To Broader Scientific Literature:**

The paper contributes to the field by demonstrating the effectiveness of SAM in enhancing the robustness of LLMs against relearning attacks during the unlearning process.

**Theoretical Claims:**

No theoretical claims.

---

> ### Author Rebuttal · Authors · 2025-03-31
>
> Thank you very much for the positive review. Your comment regarding the lack of theoretical claims has encouraged us to reflect on whether rigorous guarantees can be established to support the improved unlearning robustness enabled by SAM. While our strong empirical validation has already been acknowledged by reviewers, we agree that exploring theoretical underpinnings remains an important and valuable future direction.
>
> Inspired by the comment, we made an effort to bound the least number of relearning steps against an unlearned model and link it  with the smoothness of the loss landscape, quantified by the largest eigenvalue of the Hessian matrix. We believe this is a promising and feasible direction, but it requires more substantial and rigorous theoretical development.  The possible proof is conceptually below: we leverage gradient unrolling to contrast the relearning and unlearning dynamics. Specifically, we connect the number of required relearning steps to the largest eigenvalue of the Hessian, obtained from a local quadratic approximation of the forget loss (via Taylor expansion) around the pretrained model state. This approximation enables us to characterize how SAM-induced smoothing, reflected in a reduced Hessian spectrum, influences the model's sensitivity to relearning. It could theoretically justify the enhanced robustness of SAM-based unlearning, as evidenced by the increased number of relearning steps required to reverse the forget loss (directly linked to the Hessian spectrum).
>
> We will make our best effort to complete the proof and include it in the revised version if successful. If not, we will clearly outline this as future work and discuss the associated challenges in detail.

---

### Official Review · Reviewer_RRKK · 2025-03-14

**Overall Recommendation:** 3

**Summary:**

This paper investigates improving the robustness of LLM unlearning against relearning attacks by incorporating sharpness-aware minimization (SAM) and other smoothness optimization techniques. The authors draw an analogy between robust unlearning and adversarial training, formulating the problem as a min-max optimization task. Experiments on WMDP and MUSE datasets demonstrate that SAM and other smoothness-promoting methods improve resistance to relearning attacks and even provide some robustness against jailbreaking attacks.

**Claims And Evidence:**

The authors claim that SAM provides a robust optimization framework for LLM unlearning and significantly improves resilience to relearning attacks. While the empirical results support the claim that smoothness optimization enhances robustness, the paper lacks a detailed computational efficiency analysis, which is crucial given that techniques like SAM introduce additional overhead. Additionally, SAM is a well-established technique with broad applications, and its role in unlearning seems more like an adaptation rather than a fundamentally new algorithmic contribution. The marginal gains of SAM over other smoothness techniques, as seen in Table 1, also raise questions about its distinct effectiveness. A more in-depth theoretical justification or additional comparative studies would strengthen the claims.

**Essential References Not Discussed:**

The paper cites most of the relevant prior work, including influence function-based and knowledge attribution-based unlearning methods. However, these approaches are not deeply discussed or analyzed in comparison to the proposed smoothness-based method.

**Experimental Designs Or Analyses:**

The paper does not provide runtime or computational cost comparisons across different techniques, which is important given that SAM introduces additional optimization steps.

**Methods And Evaluation Criteria:**

The proposed methodology is reasonable, leveraging smoothness optimization to mitigate relearning attacks. However, the study does not analyze the computational cost associated with different smoothness techniques, which is critical for real-world applications. While SAM is emphasized as the core contribution, the performance difference between SAM and other smoothness techniques (e.g., gradient penalties, weight averaging) appears relatively small in Table 1, raising questions about its necessity as the primary approach.

**Other Comments Or Suggestions:**

N/A

**Other Strengths And Weaknesses:**

Strengths:
1. The paper is well-organized and easy to follow.
2. The experimental setup and evaluation methodology are clearly described.
3. The study provides an interesting connection between SAM and robust unlearning, which could inspire further research in this direction.

Weaknesses:
1. Computational overhead of different techniques is not analyzed.
2. SAM’s marginal advantage over other smoothness techniques is not well justified (as seen in Table 1).
3. No theoretical insights are provided beyond the empirical observations.
4. Lack of broader comparisons with other unlearning paradigms beyond smoothness optimization.

**Questions For Authors:**

How does the computational cost of SAM compare to other smoothness optimization techniques? Does it introduce significant overhead?

**Relation To Broader Scientific Literature:**

The paper references influence function-based and knowledge attribution-based unlearning approaches and includes NPO as a baseline. However, the evaluation primarily focuses on smoothness optimization techniques rather than a direct comparison with alternative unlearning frameworks.

**Theoretical Claims:**

N/A

---

> ### Author Rebuttal · Authors · 2025-03-31
>
> We sincerely thank Reviewer RRKK for the thorough and thoughtful review.  Below, we address each key point raised in the comments.
>
> 1. **Response to Computation Efficiency**
>
> Thank you for your constructive feedback. **[Fig. R1](https://ibb.co/4Rqfnq8d)** presents the total run time of our proposed smoothness-enhanced unlearning approachs as well as the additional baseline approach, **TAR** (Tampering Attack Resistance via meta-learning) [1].
>
> As we can see, NPO+SAM shows the second-best efficiency, slightly behind NPO+WA. It approximately doubles the runtime of vanilla NPO due to the one-step maximization for weight perturbation in the alternating gradient ascent-descent implementation of SAM. In addition, TAR incurs a prohibitively high computational cost, with a running time of 7,441.9 minutes, which is 647× slower than NPO+SAM.
>
>
> 2. **Resposne to simple adaptation of SAM and small marginal gainsover other smoothness techniques**
>
> First, we respectfully clarify that our work is not a simple adaptation of SAM, and establishing connections to other smoothness techniques is an essential contribution of our paper, as noted by Reviewer ZQJD: "First work to connect SAM and smoothness optimization to LLM unlearning robustness.", and Reviewer s57: “The analogy between unlearning robustness and adversarial training is insightful and well-justified. Introducing SAM-based optimization as a solution is a novel contribution to the field.” and “Derives the connection between curvature regularization and robustness against relearning.”.
> Second, we do not think that the performance gains of SAM are marginal. As shown in Table 1, NPO+SAM consistently outperforms the second-best smoothness method, though the runner-up may vary across different relearning settings. For instance, while NPO+SAM and NPO+RS perform similarly at $N=40$ with UE = 0.5, the performance gap widens at $M=3$, where NPO+SAM achieves UE = 0.59 versus NPO+RS at UE = 0.42. Moreover, Table 6 demonstrates that NPO+SAM also provides consistent improvements in robustness against jailbreaking attacks.
>
> 3. **Regarding theoretical explanation**
>
> This is a very intriguing comment. During the rebuttal, we made an effort to bound the least number of relearning steps against an unlearned model and link it  with the smoothness of the loss landscape, quantified by the largest eigenvalue of the Hessian matrix. We believe this is a promising and feasible direction, but it requires more substantial and rigorous theoretical development.  The possible proof is conceptually below: we leverage gradient unrolling to contrast the relearning and unlearning dynamics. Specifically, we connect the number of required relearning steps to the largest eigenvalue of the Hessian, obtained from a local quadratic approximation of the forget loss (via Taylor expansion) around the pretrained model state. This approximation enables us to characterize how SAM-induced smoothing, reflected in a reduced Hessian spectrum, influences the model's sensitivity to relearning. It could theoretically justify the enhanced robustness of SAM-based unlearning, as evidenced by the increased number of relearning steps required to reverse the forget loss (directly linked to the Hessian spectrum).
>
> We will make our best effort to complete the proof and include it in the revised version if successful. If not, we will clearly outline this as future work and discuss the associated challenges in detail.
>
>
> 4. **Response to More Robust Unlearning Methods**
>
> Thank you for your valuable suggestion. To address this concern, we conducted additional experiments comparing our proposed SAM-based unlearning method with two new baselines: TAR (Tampering Attack Resistance via meta-learning) [1] and LAT (Latent Adversarial Training, which enhances robustness through neuron activation perturbations) [2]. The results, shown in **[Table R1](https://ibb.co/S7fFVRfX)**, demonstrate that NPO+SAM achieves highly competitive performance on the WMDP-Bio unlearning task, comparable to TAR and significantly outperforming LAT. The clear advantage of NPO+SAM over LAT highlights the importance of weight-space perturbations (as in SAM) over activation-space perturbations (as in LAT). As noted in Line 77, TAR formulates unlearning vs. relearning as a meta-learning problem. However, computing the meta-gradient requires a series of gradient unrolling steps, resulting in extreme computational overhead. As we responded in the 1st question and shown in **[Fig. R1](https://ibb.co/4Rqfnq8d)**, TAR incurs a running time of 7,441.9 minutes, making it 647× slower than NPO+SAM, and thus impractical for large-scale LLM unlearning.
>
> [1] Tamirisa, Rishub, et al. "Tamper-resistant safeguards for open-weight llms." arXiv preprint arXiv:2408.00761 (2024).
>
> [2] Sheshadri, Abhay, et al. "Latent adversarial training improves robustness to persistent harmful behaviors in llms." arXiv preprint arXiv:2407.15549 (2024).

---

### Official Review · Reviewer_ZQJD · 2025-03-19

**Overall Recommendation:** 5

**Summary:**

This paper addresses the challenge of robust LLM unlearning, where undesired knowledge is removed from a large language model (LLM) without requiring full retraining. A key issue with existing unlearning methods is their vulnerability to relearning attacks, where a small fine-tuning step can restore forgotten information. The paper draws an analogy between relearning attacks and adversarial attacks, proposing Sharpness-Aware Minimization (SAM) as a solution to improve the robustness of LLM unlearning.

The key contributions of this paper are:

- Establishing SAM as a robust optimization foundation for resisting relearning attacks.
- Extending beyond SAM by exploring other smoothness optimization techniques (Randomized Smoothing, Gradient Penalty, Curvature Regularization, and Weight Averaging).
- Conducting experiments on WMDP and MUSE datasets, demonstrating that SAM-based unlearning is significantly more resistant to relearning and jailbreaking attacks.
- Providing loss landscape visualizations that show how smoothness optimization flattens the loss surface, improving unlearning stability.

The results indicate that SAM-enhanced unlearning consistently outperforms state-of-the-art methods in resisting both relearning and adversarial jailbreaking attacks.

**Claims And Evidence:**

The paper makes several strong claims regarding the effectiveness of SAM and smoothness optimization for robust LLM unlearning.

Supported Claims:

- Relearning attacks can reverse unlearning effects → Verified experimentally on WMDP/MUSE datasets.
- SAM minimizes relearning vulnerability → Shown through min-max optimization formulation, experimental results, and loss landscape analysis.
- Other smoothness techniques (RS, GP, CR, WA) also improve robustness → Experiments demonstrate that all variants improve resilience compared to the baseline.
- SAM-based unlearning is also resistant to jailbreaking attacks → The authors show improved KL divergence on adversarial prompts.

Weak Claims:

- The claim that SAM is the "optimal" robust unlearning method may be too strong. While it performs the best in their experiments, there could be alternative robustness strategies (e.g., meta-learning-based unlearning, Bayesian methods) that were not explored.
- The paper does not provide an in-depth theoretical explanation of why SAM is superior beyond empirical results. Some theoretical insights on why SAM discourages relearning in LLMs specifically could strengthen the claim.

**Essential References Not Discussed:**

NA

**Experimental Designs Or Analyses:**

- The WMDP and MUSE datasets are appropriate for testing unlearning and relearning vulnerabilities.
- The experiments are well-structured: They test the effect of smoothness-enhanced unlearning against different relearning attack intensities (epochs, sample sizes).
- The loss landscape visualizations clearly demonstrate how SAM and other smoothness methods flatten the loss surface.

Potential issues:

- The paper primarily focuses on Zephyr-7B-beta and LLaMA-2-7B, which are relatively small-scale models compared to cutting-edge LLMs (e.g., GPT-4, LLaMA-3). Would these results generalize to much larger models?
- The experiments use only one unlearning baseline per dataset (NPO for MUSE, NPO/GradDiff/RMU for WMDP). Adding other methods (e.g., model editing, knowledge distillation-based unlearning) could provide a broader comparison.
- Ablation studies on the hyperparameters ($\rho$ in SAM, number of smoothness layers in RMU) are useful, but further sensitivity analysis would be beneficial.

**Methods And Evaluation Criteria:**

- The benchmark datasets (WMDP and MUSE) are well-chosen and represent real-world unlearning scenarios, including hazardous content removal and copyrighted data forgetting.
- The paper evaluates multiple unlearning methods (NPO, GradDiff, RMU) and compares them with smoothness-enhanced versions, providing a fair comparison.
- Unlearning robustness is assessed via multiple metrics (Unlearning Effectiveness, Utility Retention, Relearning Resistance, Jailbreaking Robustness).

**Other Comments Or Suggestions:**

- Clarify whether SAM slows down LLM inference/training significantly.
- Provide examples where SAM fails to prevent relearning (e.g., highly structured knowledge).
- Test SAM-based unlearning on larger models like LLaMA-3 or GPT-4 to validate scalability.

**Other Strengths And Weaknesses:**

Strengths:

- First work to connect SAM and smoothness optimization to LLM unlearning robustness.
- The paper is well-written, with clear motivation, theoretical insights, and experiments.
- Could improve AI safety, privacy, and compliance with legal regulations (e.g., GDPR, right to be forgotten).

Weaknesses:

- The approach is computationally expensive, as SAM requires perturbation-based training, increasing cost.
- Does not address whether LLM unlearning itself could be adversarially misused (e.g., selectively erasing safety mechanisms).

**Questions For Authors:**

1. How does SAM compare to alternative robust unlearning methods (e.g., meta-learning or Bayesian forgetting)?
2. Would different fine-tuning methods for relearning (e.g., RLHF, LoRA) change the attack effectiveness?
3. Does SAM negatively impact generalization or increase catastrophic forgetting on retain data?

**Relation To Broader Scientific Literature:**

- The paper extends research on LLM unlearning (Yao et al., 2024; Maini et al., 2024) by proposing a robust optimization framework using SAM.
- It connects adversarial robustness techniques (Madry et al., 2018; Foret et al., 2021) to the field of unlearning, which has not been widely explored before.
- The findings align with prior work on curvature-based regularization (Moosavi-Dezfooli et al., 2019) and weight smoothing (Izmailov et al., 2018).

**Theoretical Claims:**

- The paper builds on SAM-based optimization and adapts it to LLM unlearning, using min-max optimization to counter relearning attacks.
- Derivations of the SAM-enhanced loss function (Equation 3-7) appear correct and align with sharpness-aware training literature.
- The connection between SAM and curvature regularization is well-explained, but the paper does not rigorously prove that SAM minimizes relearning risk optimally.

No major theoretical errors were found, but a more formal generalization bound on SAM’s effect on unlearning would strengthen the work.

---

> ### Author Rebuttal · Authors · 2025-03-31
>
> We thank Reviewer ZQJD for the thorough review and the encouraging comments on our contributions and presentation. We also greatly appreciate the constructive feedback. Below, we address each key point raised in the comments.
>
> 1.**Regarding more robust unlearning methods and larger model evaluation**
>
> Thank you for your valuable suggestion. To address this concern, we conducted additional experiments to compare our proposed SAM-based unlearning method with two additional baselines: **TAR** (Tampering Attack Resistance via meta-learning) [1] and **LAT** (Latent Adversarial Training, which enhances robustness by adding perturbations to neuron activations) [2]. These experiments were performed using a **larger model, LLaMA 3 8B**, which is the largest model supported in our lab environment. The detailed results are presented in **[Table R1](https://ibb.co/S7fFVRfX)**. As we can see, NPO+SAM delivers highly competitive performance on the WMDP-Bio unlearning task, comparable to TAR and significantly outperforming LAT. Notably, TAR incurs a prohibitively high computational cost, with a running time of 7,441.9 minutes, which is 647× slower than NPO+SAM.
>
> 2. **Regarding computation efficiency of smoothness-enhanced NPO**
>
> Thank you for the insightful suggestion. We added experiments and clarifications on the computational efficiency of smoothness-enhanced NPO methods in **[Fig. R1](https://ibb.co/4Rqfnq8d)**, along with the comparison to TAR in the earlier response. NPO+SAM shows the second-best efficiency, slightly behind NPO+WA, while offering the strongest robustness against both relearning and jailbreaking attacks. It approximately doubles the runtime of vanilla NPO due to the one-step maximization for weight perturbation in the alternating gradient ascent-descent implementation of SAM (Appendix A).
>
> 3. **Regarding theoretical explanation**
>
> This is a very intriguing comment and has prompted us to consider whether any rigorous guarantees can be provided to support the improved unlearning robustness achieved by SAM.
> Inspired by this, we made an effort to bound the least number of relearning steps against an unlearned model and link it  with the smoothness of the loss landscape, quantified by the largest eigenvalue of the Hessian matrix. Due to space constraints, please refer to **[Response to Reviewer hvpj](https://openreview.net/forum?id=zZjLv6F0Ks&noteId=Mcy2h7DqNX)** for more details.
>
> 4. **Regarding different fine-tuning methods for relearning**
>
> Following the suggestion, we perform the relearning attack using LoRA. The results are shown in **[Table R2](https://ibb.co/v4MvfCgt)**. As observed, NPO+SAM achieves a much higher unlearning effectiveness after the relearning attack compared to NPO, indicating that our method is effective not only against full-model relearning attacks but also against LoRA-based relearning attacks.
>
> 5. **Regarding the impact on retain data**
>
> In our experiments, we did not observe any clear negative impact of incorporating SAM on retain performance. For instance, MUSE uses the KnowMem metric on the retain set to directly evaluate the performance of the unlearned model on retained knowledge. As shown in Table A2, NPO+SAM achieves higher KnowMem scores than vanilla NPO, 42.58 vs. 41.58 on MUSE-News and 35.48 vs. 34.71 on MUSE-Books. This aligns with SAM’s known ability to improve generalization via loss landscape flatness. While SAM was originally designed for generalization, our contribution lies in adapting it to unlearning and showing that its generalization benefits can enhance unlearning robustness.
>
> 6. **Other suggestions/comments on sensitivity analysis, sample strategy and failure example**
>
> Thank you for acknowledging our experiments on ρ in SAM and the number of smoothness layers in RMU. In the revised version, we will include additional sensitivity analyses, such as the Gaussian noise variance $\sigma^2$ in RS, and explore variations in fine-tuning methods, including LoRA (as noted in our earlier response).
>
> Furthermore, below is an example where SAM fails to prevent relearning:
> > Prompt: What gradients are involved with scHPXV purifcation? A.Sucrose gradients; B.D-mannose gradients…
>
> > Origin model:A.Sucrose gradients\n\nExplanation:\nHorsepox virus (HPXV) is a double-stranded DNA virus…
>
> > Unlearned model before relearning: /******/ /******/ /******/…
>
> > Unlearned model after relearning (that fails to prevent relearning): A. Sucrose gradients.\nThe specific density gradient used depends on the characteristics…
>
> Third, we will improve our broader impact statement by discussing the potential misuse of unlearning techniques by adversaries to erase safety mechanisms.
>
> [1] Tamirisa, Rishub, et al. "Tamper-resistant safeguards for open-weight llms." arXiv preprint arXiv:2408.00761 (2024).
>
> [2] Sheshadri, Abhay, et al. "Latent adversarial training improves robustness to persistent harmful behaviors in llms." arXiv preprint arXiv:2407.15549 (2024).

---

### Decision · Program_Chairs · 2025-05-01

**Decision:**

Accept (poster)

**Comment:**

This paper presents a novel approach to robust LLM unlearning by leveraging Sharpness-Aware Minimization (SAM) to defend against relearning and jailbreaking attacks. Reviewers found the work to be timely and impactful, highlighting the innovative connection between loss landscape smoothness and unlearning robustness, as well as the strong empirical performance. Key concerns prior to the rebuttal included the lack of theoretical justification, evaluation on smaller models, and computational efficiency. The authors responded thoroughly, providing efficiency analysis, experiments on larger models (e.g., LLaMA-3 8B), and additional evaluations against diverse attacks. One reviewer increased their score following the rebuttal, while others acknowledged the improvements. Overall, the paper is recommended for acceptance due to its conceptual novelty, practical relevance, and thorough validation. Further theoretical development and broader attack coverage are suggested directions for future work.